# Toll-Like Receptors: Are They Taking a Toll on the Heart in Viral Myocarditis?

**DOI:** 10.3390/v13061003

**Published:** 2021-05-27

**Authors:** Kasper Favere, Matthias Bosman, Karin Klingel, Stephane Heymans, Sophie Van Linthout, Peter L. Delputte, Johan De Sutter, Hein Heidbuchel, Pieter-Jan Guns

**Affiliations:** 1Laboratory of Physiopharmacology, GENCOR, University of Antwerp, 2610 Antwerp, Belgium; matthias.bosman@uantwerpen.be (M.B.); pieter-jan.guns@uantwerpen.be (P.-J.G.); 2Research Group Cardiovascular Diseases, GENCOR, University of Antwerp, 2610 Antwerp, Belgium; Hein.Heidbuchel@uza.be; 3Department of Cardiology, Antwerp University Hospital, 2650 Antwerp, Belgium; 4Department of Internal Medicine, Ghent University, 9000 Ghent, Belgium; johan.desutter@ugent.be; 5Cardiopathology, Institute for Pathology, University Hospital Tuebingen, 72076 Tuebingen, Germany; karin.klingel@med.uni-tuebingen.de; 6Department of Cardiology, Maastricht University, 6229 ER Maastricht, The Netherlands; s.heymans@maastrichtuniversity.nl; 7Centre for Molecular and Vascular Biology, KU Leuven, 3000 Leuven, Belgium; 8BIH Center for Regenerative Therapies (BCRT), Berlin Institute of Health (BIH) at Charité, Universitätsmedizin Berlin, 10117 Berlin, Germany; sophie.van-linthout@charite.de; 9German Centre for Cardiovascular Research (DZHK), Partner Site Berlin, 10785 Berlin, Germany; 10Laboratory of Microbiology, Parasitology and Hygiene, University of Antwerp, 2610 Antwerp, Belgium; peter.delputte@uantwerpen.be

**Keywords:** viral myocarditis, Toll-like receptor, immunity, inflammation, review

## Abstract

Myocarditis is an inflammatory disease of the heart with viral infections being the most common aetiology. Its complex biology remains poorly understood and its clinical management is one of the most challenging in the field of cardiology. Toll-like receptors (TLRs), a family of evolutionarily conserved pattern recognition receptors, are increasingly known to be implicated in the pathophysiology of viral myocarditis. Their central role in innate and adaptive immune responses, and in the inflammatory reaction that ensues, indeed makes them prime candidates to profoundly affect every stage of the disease process. This review describes the pathogenesis and pathophysiology of viral myocarditis, and scrutinises the role of TLRs in every phase. We conclude with directions for future research in this field.

## 1. Introduction

Myocarditis is an inflammatory disease of the myocardium [1,2,3,4]. Although the exact aetiology often remains undetermined, a wide variety of microbial pathogens, systemic and autoimmune diseases, hypersensitivity reactions, physical noxa, drugs and toxins have been reported to cause myocarditis [2,5,6]. Among these triggers, viral infections are the most common aetiology, particularly in children [2,5,7,8]. It has been estimated that 1–5% of patients with an acute viral infection may exhibit a form of clinical or subclinical myocarditis [7,9].

The clinical presentation of viral myocarditis is highly variable, ranging from subclinical disease to fulminant myocarditis with cardiogenic shock and/or life-threatening arrhythmias [1,2,10]. Myocarditis has been identified as an important cause of sudden cardiac death, especially in young adults and athletes [11,12,13,14,15,16]. The natural history may vary from complete recovery to progressive ventricular dysfunction, eventually leading to dilated cardiomyopathy (DCM) in up to 30% of biopsy-proven myocarditis cases [2,8]. As aetiology-driven therapy is lacking, management is currently limited to supportive care for heart failure and arrhythmias [2,17]. A better understanding of the mechanisms involved may open opportunities for new, more disease-targeted therapies [18].

Many questions remain unanswered regarding the pathogenesis of viral myocarditis and the role of host immunity in disease progression and prognosis [4]. In viral myocarditis, triggering of Toll-like receptors (TLRs), a family of widely expressed pattern recognition receptors, importantly contributes to immune system activation [19,20]. These receptors recognise a variety of exogenous and endogenous signals and play a central role in both innate and adaptive immunity [21]. Accumulating evidence indicates that TLR signalling contributes to the pathogenesis of cardiac conditions in which myocardial inflammation plays a prominent role, including viral myocarditis [16,22,23].

This review aims to provide an overview of the pathogenesis and pathophysiology of viral myocarditis and the potential role of TLRs therein. Understanding the role of TLR signalling in myocarditis may help to identify diagnostic and therapeutic targets, ultimately leading to improved care of myocarditis patients [16].

## 2. Toll-Like Receptors

Toll-like receptors are a family of evolutionarily conserved type I transmembrane glycoprotein receptors [24]. To date, 13 mammalian (TLR1-13) and 10 human (TLR1-10) paralogues have been identified [25,26]. TLRs recognise distinct molecular motifs and are therefore designated as pattern recognition receptors (PRRs) [26]. These motifs can be categorised into pathogen-associated molecular patterns (PAMPs) and damage-associated molecular patterns (DAMPs). PAMPs are invariant molecular structures (inter alia (i.a.) proteins, lipids and nucleic acids) shared by a broad spectrum of microbial pathogens. As such, TLRs recognise a wide variety of viruses, bacteria, fungi and protozoa [24,27]. DAMPs, on the other hand, are endogenous signals released by stressed or injured tissues [28]. DAMPs have been subdivided into three categories, although not mutually exclusive: leaderless proteins (i.e., not containing the classical signal peptide sequences) secreted by professional immune cells (also referred to as ‘alarmins’, e.g., high mobility box group 1 protein), intracellular or surface-expressed molecules of stressed or dying cells (e.g., heat shock proteins, S100A8 or S100A9) and components of the extracellular matrix [29,30,31]. The immune response that follows DAMP recognition serves to clear debris and initiate tissue repair [32].

According to their subcellular location, the TLR family can be subdivided into two groups. TLR1, 2, 4–6 and 10 are expressed on the cell surface. In contrast, TLR3, and 7–9 reside on the membranes of intracellular compartments (such as endosomes, lysosomes, endolysosomes and endoplasmic reticulum), which permits them to recognise genetic material of intruding viruses [16,33]. For an overview of the naturally occurring and synthetic ligands of each of the TLRs, we refer to prior work of our group [34].

The key functions of TLRs in the innate and adaptive immunity run through complex intracellular signalling cascades that activate transcription factors. Figure 1 provides a schematic overview of these signalling pathways.

The signalling cascade starts with the dimerisation of the common cytoplasmic Toll/IL-1R receptor (TIR) domain [35]. Subsequently, specific adaptor proteins are recruited. Depending on the adaptor molecule recruited, TLR signalling is generally divided into two distinct pathways [36].

Myeloid differentiation primary response gene 88 (MyD88) is an adaptor protein that is utilised by all TLRs except for TLR3, and that initiates the MyD88-dependent pathway. After TLR engagement, MyD88 complexes with IL-1R-associated kinases (IRAKs), which results in the activation of TNFR-associated factor (TRAF) 6 and subsequently TGF-β-activating kinase (TAK) 1. The latter belongs to the mitogen-activated protein kinase kinase kinase (MAPKKK) family and in turn activates two different pathways. In the first pathway, TAK1 binds to the IκB kinase (IKK) complex, which subsequently leads to the phosphorylation and proteasome degradation of the NF-κB inhibitory protein IκB. Herewith, the transcription factor NF-κB is able to translocate from the cytosol into the nucleus. NF-κB stimulates a wide variety of genes, including pro-inflammatory cytokines. In the second pathway, TAK1 activates MAPK family members such as extracellular signal-regulated protein kinase (ERK) 1/2, p38 and c-Jun N-terminal kinase (JNK). These factors mediate the activation of the pro-inflammatory transcription factor activator protein-1 (AP-1) [35]. 

TLR3 (solely) and TLR4 (in addition to the MyD88-dependent pathway) can recruit the TIR-domain-containing adaptor protein-inducing interferon-β (TRIF) [16,35]. TRIF interacts with TRAF6 and TRAF3. As elaborated above, TRAF6 ultimately leads to activation of the transcription factors NF-κB and AP-1. In contrast, TRAF3 recruits TANK-binding kinase 1 (TBK1) and inducible IκB kinase IKKi (also known as IKKε) for phosphorylation of interferon regulatory transcription factors (IRFs) 3 and 7. This way, IRFs are able to translocate to the nucleus where they induce the expression of type I interferon (IFN) genes and IFN-inducible genes [16,35]. IFNs are multifunctional cytokines that play an important role in the first line defence against viral pathogens [37]. 

So ultimately, TLR activation promotes transcriptional activity of a wide variety of genes, including chemokines, type I interferons (i.a. IFN-α and IFN-β) and other cytokines. Subsequently, the innate immune system is stimulated and adaptive immune responses are primed and orchestrated [35,38].

TLRs are widely distributed in various cell types and tissues, but are primarily expressed in immune cells [39]. The TLR expression pattern is tissue-specific, including differences across immune cell subtypes [40]. Also the human heart harbours TLRs [41,42,43]. More specifically, their presence has been demonstrated in cardiomyocytes, endothelial cells, smooth muscle cells and fibroblasts [16,22,44]. The order of the relative TLR mRNA expression level in adult human heart tissue from high to low is: TLR4 >> TLR2 > TLR3 >> TLR5 > TLR 1 > TLR6 > TLR7 >> TLR8 > TLR9 > TLR10 [41]. Notably, the relative mRNA expression levels of TLR 2–4 are approximately 10-fold higher than of TLR1 and 5–10 [26,41]. Protein expression of TLR1–9 has been confirmed in human primary cardiac cells or human cardiac tissue [44,45]. To the best of our knowledge, the protein expression of TLR10, notably the sole TLR without known ligand and also the TLR with the lowest transcript level, has not yet been confirmed in the heart [46,47]. 

Interestingly, in vitro infection of human cardiac cells with coxsackievirus B3 (CVB3) increases the expression of TLR7 and TLR8 on these cells [45]. Similarly, Satoh et al. demonstrated higher TLR4 mRNA expression levels in endomyocardial biopsy samples of myocarditis patients compared to controls [48]. Also in patients with ‘idiopathic’ DCM, a potential consequence of viral myocarditis, focal areas of increased, intense TLR4 staining in the heart have been described [49].

## 3. Viral Myocarditis

In the next sections, we discuss the pathogenesis and pathophysiology of viral myocarditis and the potential role of TLRs therein. Because of the heterogeneity in clinical presentation and the difficulty of diagnosis, most of our understanding of viral myocarditis is derived from experimental (coxsackie)virus studies in mice [5,6,50]. Further in this review, we elaborate on the implications for translation of these insights to man.

A wide variety of viruses have been implicated in human myocarditis, including some of the most prevalent, seemingly harmless viral agents. Group B coxsackieviruses were the most frequently identified viruses in endomyocardial biopsies of myocarditis patients in the 1950s through 1990s. More recently, adenoviruses in children and parvovirus B19 (B19V) have been increasingly detected [51,52,53]. The reason for this shift in causative pathogens is currently unclear. Much controversy surrounds the detection of B19V in myocardial tissue as cardiac viral persistence can occur, viral copy numbers are often low and also individuals without myocarditis or DCM frequently harbour B19V DNA in their myocardium. Therefore, persistence of B19V DNA in absence of inflammation is most likely not clinically relevant [4].

The tropism of a virus for particular target tissues is a complex phenomenon that involves multiple factors. One factor are the environmental conditions (temperature, anatomical barriers and route of inoculation). Another factor is (co)receptor expression. The majority of viral agents can only enter the terminal target cells if these express an appropriate receptor. However, receptor expression can be dynamic and some viruses require several (co)receptors or utilise different receptors on different host cells in different tissues. In addition, other cellular factors such as the presence of specific transcription factors or cellular proteases (for example required for activation of virus attachment or fusion proteins) are of importance [54,55,56,57].

To cause myocarditis, the virus usually spreads to the heart via lymphatic and/or haematogenous routes [54]. Although the majority of viruses associated with myocarditis are capable of directly infecting cardiomyocytes, also other cardiac cell populations are potentially prone to infection [58]. For example, the vasculotropic B19V primarily targets intracardiac endothelial cells [59,60,61]. To date, detailed knowledge on which cell types are specifically infected by myocarditis-associated viruses remains scarce [19,58,62]. As discussed in the review by Tschöpe et al., it is important to distinguish viruses that directly infiltrate the heart (cardiotropic and vasculotropic viruses, e.g., enteroviruses), viruses that indirectly infiltrate the heart (which are generally lymphotropic, such as herpesviruses), and viruses that do not infect cardiac cells but indirectly induce cardiac damage as part of a more generalised immune system activation (for example in the context of cytokine storms, e.g., influenzaviruses) or by triggering an autoimmune response to cardiac components [4]. In the setting of COVID-19, current evidence suggests different mechanisms of myocardial damage. These include indirect mechanisms such as cytokine storm, triggering of a cardiac autoimmune response, respiratory dysfunction and decreased activity of angiotensin-converting enzyme 2 (ACE2) and the associated ACE2/angiotensin-(1–7)/MAS axis. Detection of SARS-CoV-2 viral particles in cardiac macrophages implies that the virus can reach the heart during viraemia or through infiltration of infected macrophages [63]. Although cardiomyocytes express ACE2, the host receptor for SARS-CoV-2, viral presence has not yet been demonstrated in cardiomyocytes. In contrast, endotheliitis with presence of viral elements within the endothelial cells has been reported in different organs of COVID-19 patients, including the heart [4,64,65].

The natural course of viral myocarditis is classically divided into three distinct phases, which represent an artificial subdivision of a continuum and are therefore, not surprisingly, topic of debate [2,4,7,8,58,66,67,68,69,70,71]. The first phase entails direct microbial damage. Viral replication and virus-mediated cell lysis lead to apoptosis and necrosis [2,8,67,72]. In phase II, an immunopathological state develops in which antiviral and/or autoimmune responses perpetuate the inflammatory process [67]. Phase III constitutes the process of healing and cardiac remodeling [2,8,67]. In their review on the topic, Yajima and Knowlton introduced the pre-infection phase 0. This phase mainly compromises the underlying susceptibility to the development of viral myocarditis [58]. The different myocarditis phases are summarised in Figure 2. The role of TLRs in each phase is schematically presented in Figure 3.

## 4. Phase 0—Predisposing Factors

The viruses that cause myocarditis are actually common viral agents in human infectious diseases. The lifetime risk of infection with at least one of the aforementioned viruses is estimated to be greater than 90%, if not 100% [73,74,75]. Therefore, the question remains why only a minority of virally infected patients develop myocarditis. Currently, there is no evidence to assume that these individuals are generally more susceptible to other infectious diseases. This implies that host and/or viral factors specific to viral myocarditis are involved [58].

Murine models have provided strong evidence for the importance of genetic background in the development of viral myocarditis. Susceptible (e.g., BALB/c, A/J) and resistant (e.g., C57BL/6) murine strains can be distinguished. Whereas all strains initially suffer from acute myocarditis, solely susceptible strains develop chronic myocarditis as a result of virus persistence and/or perpetuating inflammation [76,77,78,79]. Factors that have been associated with these strain differences are mostly related to the immune system and include major histocompatibility complex haplotype, Th1/Th2 balance, amount of cross-priming dendritic cells and secretion of cytokines (IFN-γ, IL-10, interferon-inducible protein 10 (IP-10)) [80,81,82].

Also in man, several host factors have already been put forward. These include genetic factors involving the major histocompatibility complex class II and the CC-chemokine receptor 5 (CCR5) [83,84,85,86,87]. In addition to immune factors, inherited deficiencies of sarcolemmal proteins (dystrophin, dysferlin) have been suggested as predisposing factors for viral propagation [7]. Interindividual differences in virus receptor expression represent another potential mechanism to explain the variability in presentation, clinical course and outcome of viral myocarditis. Coxsackie-adenovirus receptor (CAR) (the common receptor for coxsackievirus group B and adenovirus serotypes 2 and 5) overexpression indeed enhances virus uptake in vitro [88]. CAR was found to be upregulated in the myocardium of explanted DCM hearts compared to hearts of patients with other cardiac diseases or healthy donors [68,88,89]. However, it has become evident that CAR upregulation is not limited to inflammatory cardiomyopathy or DCM, but also occurs in cardiac pathology unrelated to myocarditis [90,91,92,93]. As CAR has a significant role in foetal cardiac development, upregulation might have importance in the regeneration of damaged myocardium [90,94]. This suggests that the previously injured heart may have a particular predisposition towards infection with cardiotropic viruses [94]. Curiously, transgenic mice overexpressing CAR rapidly developed cardiomyopathy, which contrasts with the aforementioned hypothesis that CAR induction acts on myocardial regeneration [95]. MyD88, the adaptor protein utilised by the majority of TLRs, is likely to be involved in the regulation of CAR expression as MyD88-deficient mice exhibit reduced CAR protein expression in the context of murine viral myocarditis [69]. It is clear that the exact role of CAR in cardiac disease states and in the susceptibility for viral myocarditis needs further elucidation.

Genetic variants of TLRs affect (viral) myocarditis susceptibility as well. In 19 patients with biopsy-proven enteroviral myocarditis, a common single nucleotide polymorphism (SNP) in the TLR3 gene was identified in 10 cases, with 4 cases (i.e., 21%) even being homozygous. In the control population, only 4% was homozygous for this SNP. Expression of the mutated TLR3 in different cell lines abrogated activation of the type I interferon pathway upon CVB3 infection, leading to increased enteroviral replication [96]. Growing evidence also suggest that SNPs within the TLR4 gene are associated with susceptibility to myocardial inflammation [23]. In fact, many studies have linked TLR polymorphisms to a wide spectrum of infectious and inflammatory diseases, including susceptibility to viral infections and autoimmune diseases. Of note, loss-of-function variants in the TLR7 gene have recently been identified in young patients with grave forms of COVID-19 requiring mechanical ventilation [27,97,98]. Besides receptor expression, also the activity of signalling pathways in cardiac myocytes, e.g., ERK-1/2 activation, may contribute to differential host susceptibility to viral myocarditis [99].

Viral myocarditis is not associated with particular race predilections [100,101]. Age and sex significantly influence myocarditis susceptibility. Certain life periods are characterised by higher incidence of viral myocarditis. More specifically, infants and young adults are most commonly affected. The underlying mechanisms have not yet been unravelled, although an age-related decline in CAR expression has been reported in rodent models [5,7,9,56,84,92,102,103,104]. Further, there are several lines of evidence that sex hormones determine the susceptibility for, and the natural course and outcome of myocarditis [5,105,106,107]. In general, female sex protects against infectious myocarditis. Pregnancy, on the other hand, confers an increased risk of myocarditis [108,109,110]. It has been demonstrated that administration of testosterone and progesterone negatively affects the course of viral myocarditis [106,110,111]. Importantly, sex differences in TLR immune cell expression levels, activation susceptibilities and signalling have been found as potential explanation for the sex bias [6,7,77,105,112,113]. In fact, the impact of sex hormones could be mediated through similar mechanisms as it for example has been demonstrated that testosterone increases TLR4 expression on mast cells and macrophages, and that estradiol suppresses increased TLR4 expression after lipopolysaccharide (LPS) (a TLR4 agonist) stimulation [105,113].

Nutrition studies have illustrated that host factors can interact with viral factors. It is generally accepted that nutritional deficiencies impair host immunity [114]. Marasmic mice and mice with selenium deficiency (analogous to Keshan disease) or vitamin E deficiency experience increased severity of coxsackievirus-mediated cardiac damage [9,115,116]. Furthermore, one study reported that host deficiency in selenium or vitamin E propagated alterations in the viral genomic composition, and subsequently viral phenotype. As such, an avirulent amyocarditic coxsackievirus strain converted into a virulent strain with the capacity to induce myocarditis in mice of normal nutriture. This suggest that nutrition can affect both host and pathogen [115].

The existence of myocarditic and amyocarditic strains of the same virus emphasises that pathogen factors are undeniably of importance [117,118]. This was further illustrated by studies of clinical isolates of CVB3 that identified mutations in the 5′ nontranslated region to be associated with cardiovirulence, viral replication and virus persistence [119,120,121].

## 5. Phase 1—Direct Microbial Damage

Direct viral effects play an important role in the development of viral myocarditis [7,9,56]. It has been clearly demonstrated that the pathogen itself causes myocardial damage in the early phase of myocarditis, prior to immune cell infiltration [56]. There are several mechanisms through which viruses cause cell damage, and some mechanisms are unique to certain viruses [57,122].

After cell entry, the virus hijacks the host cell machinery to ensure viral replication [123]. Viruses can encode proteins and proteases that interfere with, or even cleave, structural host proteins and proteins involved in the cell cycle, transcription, translation, cardiac contraction and signalling transduction [5,6,56,122,124]. Their actions serve to support the virus life cycle. For instance, disruption of cell-cell and/or cell-matrix connections facilitate viral entry [125]. One of the best studied examples is the enteroviral protease 2A, which cleaves dystrophin [7,125,126]. The action of this protease is of importance for viral propagation as mice expressing cleavage-resistant dystrophin have decreased sarcolemmal disruption and cardiac viral titers following coxsackievirus infection [126].

Together, the resulting loss of cellular homeostasis, translation shut-off, contractile dysfunction and apoptosis mediated by these virus-encoded proteins contributes to direct damage of infected cells [5,56,123,127]. In the case of cytolytic viruses (such as CVB3), these proteins are also capable of inducing cell lysis through an increase of membrane permeability or pore formation [122]. In addition to apoptosis and cytolysis, myocardial infection can also induce autophagy and necrosis [57,123,128]. Upon cell death, viral progeny are released into the environment [56,129]. In a cell culture model of coxsackievirus infection, progeny viruses were detected as soon as 9 h after cell entry [56]. It was estimated that infected myocytes survive approximately 24 h during acute replication [130].

The consequences of persistent viral cardiac infection additionally illustrate the importance of direct microbial damage. Wessely et al. reported that low-level enteroviral gene expression, similar to that observed in hearts with persistent viral infection, can induce cytopathic effects in myocyte cell cultures without generation of infectious progeny [131]. The same group used a transgenic mouse model to demonstrate that restricted replication of enteroviral genomes can induce defective excitation-contraction coupling and DCM [132]. In a murine model of chronic enteroviral heart disease, a strong correlation exists, both spatial and temporal, between viral replication and myocardial lesions [130]. In patients with LV dysfunction (including DCM patients) viral myocardial persistence is associated with progressive cardiac dysfunction, whereas viral elimination leads to functional improvement. It is not clear whether the progressive cardiac dysfunction is related to the viral presence per se, or rather the result of the evoked immune response [52,133,134].

It seems that TLR activation through viral recognition affects the extent of direct microbial damage. Fairweather et al. reported reduced levels of viral replication and myocarditis in TLR4-deficient mice [135]. Interestingly, the MyD88 pathway alters the expression of the critical CAR in the process of CVB3 infection in mice. More specifically, the cardiac protein levels of CAR were significantly reduced in MyD88-deficient mice, contributing to decreased viral entry, proliferation and injury [69].

Activation of ERK-1/2, a member of the MAPKs that is, among others, involved in MyD88-dependent TLR signalling, is necessary for CVB3 replication, whereas inhibition circumvents CVB3-induced apoptosis [56,123,136]. Early activation of ERK-1/2 may stem from TLR stimulation or interaction of CVB3 with the CAR and/or associated co-receptor. At later stages, accumulation of the active form of Ras, resulting from viral protease-mediated protein cleavage, is an additional mechanism of ERK-1/2 phosphorylation [123]. Activation of p38, another MAPK family member, seems to play an important role in coxsackievirus progeny release through regulation of apoptosis [137]. Stimulation of pre-existing signal transduction pathways by enteroviruses, in this case MAPK pathways, could be an intricate viral mechanism to propagate viral replication. In addition, MAPK pathways lead to intracellular calcium mobilisation during viral replication, contributing to destruction of infected myocytes [59]. While TLRs also trigger these MAPK family members via the MyD88 universal adaptor protein, it remains unclear whether TLR activation might inadvertently contribute to viral propagation.

As mentioned earlier in this section, viral infections induce autophagy. Next to its fundamentality for cellular homeostasis, autophagy exerts important functions in innate and adaptive immunity [138,139]. One of these functions is the lysosomal degradation of microorganisms (such as viruses) that invade intracellularly [138]. However, several viruses are capable of reducing the host autophagic flux and even have subverted this host mechanism to the benefit of their own replication [138,140]. By induction of autophagosome formation but inhibition of the subsequent fusion with lysosomes, autophagosomes accumulate and provide a scaffold for the replication complex [140,141]. TLR stimulation can induce autophagy and therefore propagate viral replication, at least in a certain proportion of the viruses [139,140].

## 6. Phase 2—Pathogen-Immune System Interplay

Viral infection of the heart and release of DAMPs trigger the activation of the host immunity [7,142]. Cardiac resident cells, including myocytes, fibroblasts, endothelial cells, dendritic cells and mast cells, initiate the innate immune response through cytokine production (i.a. IL-1, IL-6, TNF-α and interferons) [43,50,56,143]. Together with the release of progeny virus into the interstitium, these local signals activate resident macrophages, the most abundant leukocyte species in the heart [3,50,142]. The production of cytokines (among which are chemokines) and the induction of endothelial adhesion molecules and chemokine receptors generates a strong inflammatory response [50,77]. The first wave of infiltrating immune cells mainly consist of natural killer (NK) cells, which selectively eliminate virally infected cells [3,6,56,58,144,145]. Monocyte recruitment translates into the presence of M1 (classically activated) macrophages, which phagocytose infected cardiomyocytes. The innate immune cells release additional cytokines and enzymes, amplifying the inflammatory process [142].

The bone marrow and splenic reservoirs of stem cells and progenitors are stimulated to recruit immune cells to the inflammatory site [142]. CVB3 is capable of infecting monocytes and the spleen is among the target organs of infection. Therefore, in CVB3-induced myocarditis, homing of immune cells from the spleen to the heart (the cardiosplenic axis) can further contribute to cardiac viral infection [4,146]. This illustrates the strong interconnection between the different myocarditis phases.

The major coordinating steps in the innate immune response involve interactions of viral PAMPs and DAMPS with the TLRs [66,94]. TLR3, 4 and 7–10, of which only TLR4 and TLR10 are expressed on the cell surface, have viruses among their PAMP-related pathogens [34]. CVB3 infection in mice induces upregulation of all TLRs, although the increase is more pronounced for those that are predominantly located in intracellular compartments [76,147]. This is most likely related to cardiac inflammation rather than viral presence per se, as upregulation of intracellular TLRs has also been described in the setting of cardiac injury of non-viral origin (e.g., acute myocardial infarction or autoimmune myocarditis) [44,148]. Activation of TLRs triggers the production of several cytokines which, as described above, recruit immune cells to the myocardium and amplify the inflammatory response [56].

In murine models, the adaptive immunity becomes fully active after 7 days [6,56,58]. As such, a wave of infiltrating antigen-specific T-lymphocytes develops, whereas the response of the specific B-lymphocytes is more delayed with a gradual increase [56,85,149]. Both T- and B-lymphocytes play a major role in limiting viral propagation [50]. Cytotoxic T-lymphocytes lyse virus-infected cardiomyocytes [3]. T helper cells do not directly kill infected cells, but are important mediators between antigen-presenting cells, cytotoxic T-lymphocytes and B-cells. B-lymphocytes produce neutralising antibodies to enhance viral clearing [3,149].

In the setting of experimentally induced viral myocarditis in mice, knockout of MyD88 led to significant reduction of p56^lck^ expression, a molecule important in T cell activation [69,150]. This observation suggests a connection between the MyD88-dependent innate immune system activation (potentially TLR-mediated) and subsequent stimulation of the adaptive immunity [69].

The immune response functions as a double-edged sword. The initial response is appropriate and imperative to combat the viral threat. In most patients with viral myocarditis, the immune response declines with virus elimination, and left ventricular (LV) function recovers without sequelae [14]. Interestingly, the intensity of the immune response elicited does not necessarily equate to viral presence [50]. Prolonged or excessive immune system activation generates additional myocardial damage [7,77]. Infiltrating immune cells targeting virus-infected cardiomyocytes may cause collateral damage to remote uninfected areas. Inappropriate activation of TLRs (for example by self-components) can result in sterile inflammation and potentially autoimmunity [29,148]. One self-component of particular interest in this setting is cardiac myosin. Its ability to directly activate TLR2 and TLR8 is probably related to structural similarity to certain PAMPs. In patients with acute myocarditis or DCM, myocarditis-derived monocytes were strongly responsive to cardiac myosin peptide TLR ligands, producing (detrimental) Th17-type cytokines [151,152].

Tissue damage incurred during the initial contact with the pathogen leads to exposure of cryptic antigens (e.g., myosin-derived peptides), which can eventually generate an autoimmune cardiac-specific response (i.e., development of autoreactive immune cells or autoantibodies) given the pro-inflammatory micro-environment in which they are released. This mechanism has been referred to as the ‘bystander effect’ or ‘epitope spreading’. In addition, the heart may also sustain damage through molecular mimicry between cardiac epitopes and virus particles. For example, antibodies against cardiac myosin and the cardiomyocyte sarcolemma have been shown to cross-react with coxsackie B viruses. A particular kind of molecular mimicry is that autoantibodies generated against myosin-derived peptides may cross-react with adrenergic receptors and exert an activating adrenergic effect upon ligation, contributing to chronic sympathic-mediated cardiac damage [6,56,122,148,153].

Notably, myocardial infection by specific microbes seems to play the role of ‘coadjuvant’ to the autoimmune adaptive response. In 2014, Noel Rose even considered myocarditis to represent the paradigm of infection-induced autoimmune disease. This coadjuvant role presumably runs through innate immunity activation, potentially via TLRs [6,154]. So it appears that the nature of the innate immune response is important in determining whether autoimmune disease develops following infection [154]. In addition, also B-cells express TLRs on their cell surface. TLR-signalling in B-cells is critically associated to B-cell activation and tolerance, and to pathological conditions such as viral myocarditis, including involvement in the generation of autoreactive plasma cells [155].

### Chronic Myocarditis & Viral Persistence

Both persistent viral infection and autoimmune injury can underlie the development of chronic myocarditis (with possible progression to DCM) [84]. Although the host defence normally functions to clear the virus, the balance between viral clearance and myocyte damage, and the temporal relationship between viral clearance and persisting immunologic surveillance may tip towards inefficient viral clearing or overaggressive immunologic activation [3].

Replicating or nonreplicating virus or viral nucleic acid can persist in the myocardium, perpetuating cardiac injury [77]. Sustained presence of viral nucleic acid has been demonstrated in the myocardium of susceptible mice strains and in human myocarditis patients [56,79,130,156]. The unexpectedly high prevalence rates of viral material in the myocardium of ‘idiopathic’ DCM patients and the fact that this seems related to the LV function in these patients, suggest a possible unrecognised viral origin [19,51,52,130,133,157].

Persistently infected myocardial cells are found primarily within foci of chronic myocardial lesions, characterised by replacement fibrosis, degenerated myocytes and mononuclear cell infiltration. Regarding the molecular mechanisms of coxsackievirus persistence, virus replication is restricted at the level of viral RNA synthesis during ongoing myocarditis. Therefore, compared to the acute phase, the number of infected cells are reduced and the copy numbers of viral genomes are decreased [59,130]. In 1999, Tam and Messner reported that coxsackievirus persistence in myofibers is facilitated through production of stable double-stranded RNA [158]. Double-stranded RNA may be associated with reduced viral antigen expression, minimising immunogenicity [159]. One year later, in viral myocarditis in a susceptible mouse strain, a biphasic pattern of virus clearance was demonstrated, with a much slower rate of decline in viral RNA levels during the post-acute phase. Remarkably, compared to non-cardiac organs (i.a. pancreas, liver and spleen), the myocardium shows longer duration of viral persistence [135,159]. On the other hand, it is believed that persistent infection of lymphoid cells (in spleen and lymph nodes) may play an important role in the maintenance of a non-cardiac viral reservoir [130,156,160].

## 7. Phase 3—Cardiac Repair and Remodelling

Acute cardiac stress or injury goes through a sequence of phases: first a pro-inflammatory stage, followed by healing, and finally remodeling [149]. The latter term refers to the robust plasticity response of the heart and includes complex and interrelated transcriptional, signalling, structural, electrophysiological and functional alterations [161,162].

In all pathologies, cessation of myocardial inflammation requires anti-inflammatory stimuli [76]. In the absence of pro-inflammatory triggers, anti-inflammatory cytokines (e.g., TGF-β and IL-10) promote resolution of the inflammatory response. These anti-inflammatory signals are secreted by regulatory T cells (Tregs) and alternatively activated (M2) macrophages [19,77]. Correspondingly, Becher et al. demonstrated (almost complete) normalisation of the initially increased cardiac TLR expression levels 28 days after CVB3 infection in C57BL/6 mice [44].

The anti-inflammatory stimuli also initiate pro-fibrotic signaling [29,76,163]. TGF-β signals to fibroblasts, leading to chemotaxis and differentiation into myofibroblasts. During the differentiation process, the fibroblast undergoes a phenotypic switch, adopting some contractile properties to maintain tensile strength. A provisional interstitial matrix is produced by the fibroblasts (mainly producing collagen types I and III), but also by macrophages (predominantly producing collagen IV) to preserve structural integrity. Finally, the newly formed extracellular matrix (ECM) crosslinks to form a permanent scar [50].

Virus-induced necrosis seems to initiate both reparative (replacement) and reactive interstitial fibrosis [62,164]. In the former, scar tissue replaces regions of dead myocytes [29,162]. The latter is an epiphenomenon of sustained suppression of non-circumscribed self-perpetuating inflammation with activation of pro-fibrotic signals [76].

Myocardial fibrosis is a hallmark feature of ventricular remodeling [56,162,165]. Fibrosis promotes contractile dysfunction and provides a substrate for arrhythmias, hereby contributing to morbidity and mortality [162,165]. Long-term follow-up of patients with biopsy-proven viral myocarditis has identified myocardial fibrosis as the strongest independent predictor of arrhythmias and mortality [166]. Data from experimental autoimmune myocarditis have shown that selective stimulation of TLR2, TLR4 or TLR9 starting at day 10 postimmunisation increases myocardial fibrosis at day 34 [167]. To the best of our knowledge, no information is available on the impact of TLRs on myocardial fibrosis in the setting of viral myocarditis.

A specialised matrix is deposited at sites of tissue damage, comprising ECM constituents that are not expressed (or at least not at high levels) in healthy tissue. As described in the section on DAMPs, a number of these constituents can act as endogenous ligands for TLRs, directly driving sterile inflammation from within this microenvironment. Emerging data also indicate that TLR activity impacts the ECM, creating a feedback loop that synergistically controls the inflammatory microenvironment. ECM activation of TLRs also drives other outputs than merely sterile inflammation. For example, ECM glycoproteins have been shown to stimulate foam cell formation, smooth muscle activation, renal fibroblast proliferation, collagen production and myofibroblast differentiation in a TLR-dependent fashion. The proteoglycans biglycan and decorin are among the best characterised extracellular DAMPs orchestrating tissue inflammation. Current evidence supports their potential involvement in cardiac inflammation, fibrosis and remodeling [32]. Popovic et al. demonstrated that biglycan triggers autoimmune myocarditis via TLR4-mediated stimulation of cardiac peptide presentation, but in viral myocarditis, more research needs to be conducted [168].

Recent evidence has revealed decisive roles for resident and recruited innate immune cells in the coordination of repair following a cardiac insult [29,142]. Reciprocal analyses in zebrafish and medaka (rice fish) revealed that an early acute inflammatory phase promotes the regenerative response. Zebrafish harbour unique regenerative capacity, also in the heart, and this ability is not shared by the medaka. In comparison to the latter, zebrafish exhibit earlier and more pronounced macrophage recruitment upon cardiac injury. Experimentally delaying macrophage recruitment in zebrafish compromised cardiomyocyte proliferation and scar resolution. In contrast, stimulation of TLR signalling using the synthetic TLR3 ligand polyinosinic:polycytidylic acid (poly(I:C)) enhanced cardiomyocyte proliferation and scar resolution in medaka [142]. These findings support the current paradigm that the inflammatory response following injury is required for proper tissue repair. However, if this response becomes dysregulated, it can generate additional collateral damage and pathological tissue fibrosis [29]. For example, co-culture experiments have demonstrated that cardiac fibroblasts exposed to sustained inflammatory signalling exhibit an enhanced repertoire of profibrotic phenotypic responses to mast cell mediators (with presumably an important role for TGF-β) [169]. The same authors later demonstrated that necrotic myocardial cell (NMC) supernatants (containing a multitude of DAMPs) are sufficient to provoke fibroblast activation, proliferation, migration, α-SMA expression and collagen production. Treating the fibroblast cultures with neutralising TLR4 antibodies significantly diminished cell migration and proliferation. Furthermore, injecting NMC supernatants into the heart provoked increased myocardial inflammation and fibrosis in wild-type mice, but not in TLR4-deficient mice [170].

### Cardiac Dysfunction

There is solid body of evidence that cardiac remodelling is an important pathogenic factor for cardiac dysfunction post viral myocarditis [76,171,172]. Besides deterioration of the cardiac extracellular matrix, cardiomyocyte loss also contributes to the cardiac dysfunction that potentially ensues [173]. Depending on the extent of the cardiac damage, loss of contractility and ventricular enlargement may eventually result in DCM [19,77,173,174]. Thus, both innate and acquired immune responses, originally initiated to eradicate the insulting agent, may in fact contribute to the development of heart failure by inflicting tissue damage [56]. On the other hand, failure to completely eradicate the virus from the heart may result in chronic inflammation and accelerated progression to DCM [77].

Cleavage of dystrophin represents an additional mechanism by which enteroviral infection contributes to the pathogenesis of acquired forms of DCM [56,124]. The group of Kirk Knowlton demonstrated that cardiac-restricted expression of enteroviral protease 2A alone is sufficient to induce DCM [175]. Subsequently, experimental induction of a mutation at the dystrophin cleavage site largely prevented induction of cardiomyopathy by protease 2A expression [126]. The detrimental impact of dystrophin cleavage does not come as a surprise. Dystrophinopathy is an X-linked disorder caused by mutations in the DMD gene encoding for dystrophin, with Duchenne and Becker muscular dystrophy being the best known examples [125,176]. Patients suffering from this disorder often die from the early development of a hereditary X-linked DCM [176,177].

Growing evidence supports the concept that host immunity is implicated in the pathogenesis of heart failure, irrespective of the heart failure type. Especially innate immune activation seems a key pathogenic mechanism in heart failure [22,44,178]. For instance, Mann demonstrated that the expression profile of the innate immune genes in explanted hearts from heart failure patients was distinct from that of ‘non-failing hearts’. Interestingly, the expression profile in hearts of viral cardiomyopathy patients overlapped with those of idiopathic DCM patients and differed from heart failure of ischemic origin [26].

More and more emerging data indicate that TLRs and associated signalling pathway molecules are involved in the pathogenesis and progression of heart failure [22,178]. In general, TLRs are upregulated in the failing heart [22,49]. Becher et al. demonstrated that induction of TLRs clearly differed between various experimental murine heart failure models [44].

In heart failure of ischemic origin in a rat model, next to increased mRNA and protein expression, also altered TLR4 ligand-binding capacity and inflammatory function were demonstrated [179]. Importantly, heart failure is associated with increased levels of certain DAMPs (*i.a.* heat-shock proteins and reactive oxygen species). In ischemic heart failure and anthracycline-induced cardiotoxicity, the involvement of TLRs in negative cardiac remodelling has been undeniably demonstrated. In enterovirus-positive DCM patients, elevated myocardial TLR8 and MyD88 levels were recorded, and these were localised to cardiac myocytes. After a mean follow-up of more than 1 year, TLR8 levels were a strong predictor of the occurrence of clinical outcomes [180]. In the setting of viral myocarditis, further evidence is currently lacking.

The mechanisms through which TLRs and associated signalling molecules impact the remodelling process remain to be elucidated. Amongst others, modulation of inflammation, immune cell recruitment, myocardial fibrogenesis, the renin-angiotensin system (RAS), autonomic cardiac regulation and modulation of the autophagic flux have been put forward [22,181,182].

## 8. The Role of Toll-Like Receptors

Table 1 gives an overview of the results of animal studies investigating the role of TLR signalling in viral myocarditis. In all cases, knockout models were used to obtain further insights. As could be expected, the majority of studies have focused on TLRs with an intracellular location (TLR3, 7–9). The interpretation of the results is complicated by the use of mice with different backgrounds and inoculation with different viruses. Coxsackieviruses belong to the *Enterovirus* genus of single-stranded non-enveloped RNA viruses and have a direct tropism for cardiomyocytes through the CAR [183]. The murine cytomegalovirus (MCMV) is a member of the β subfamily of herpesviruses and thus has a double-stranded DNA genome [184]. In contrast to enteroviruses, MCMV cannot infect myocytes [19]. Different inbred mouse strains have different abilities to control MCMV infection, with the BALB/c strain being relatively susceptible for example [184]. After infection, the virus becomes latent in the heart of susceptible strains [185]. A detailed comparison between the type 3 coxsackievirus and MCMV can be found elsewhere [186].

In humans, we only have indirect evidence derived from association studies. Satoh et al. demonstrated that myocarditis patients exhibited higher TLR4 mRNA expression levels than controls, and increased TLR4 levels were associated with enteroviral replication and cardiac dysfunction [48].

Figure 4 depicts the proposed roles of each of the individual TLRs in viral myocarditis.

### 8.1. TLR2

The limited evidence available does not allow to define the role of TLR2 in viral myocarditis. Female mice, which show higher levels of TLR2 expression, develop less severe myocarditis and have higher survival rates than their male counterparts. Importantly, stimulation of TLR2 by Pam3CSK4 (a synthetic triacylated lipopeptide) significantly reduced mortality in male mice (whereas no effect was observed in female mice) [113]. However, the same group later demonstrated that TLR2 knockout improved survival (significantly in female mice and a trend towards significance in male mice) [187].

### 8.2. TLR3

In general, deficiency of TLR3 is associated with increased viral replication, more extensive cardiac inflammatory lesions on histopathology and increased mortality [82]. Hardarson et al. did report increased viral presence, cardiac troponin and mortality in TLR3-knockout mice (even in heterozygote), but the inflammatory scores were lower on day 3 and 5. The authors concluded that significant direct virus-induced damage had occurred in the knockout mice due to impaired cellular immune infiltration [188]. Although Abston et al. observed increased viral replication, cardiac immune cell infiltration and impaired cardiac function, mortality did not differ between TLR-deficient and wild-type animals (after a follow-up of 35 days after inoculation) [190]. Solely Pagni et al. did not find differences in myocardial viral levels compared to wild-type infected animals. Also cardiac inflammation was comparable in their model [148].

It has been suggested that the beneficial effects of TLR signalling could be mediated through type I interferons [70]. Dendritic cells of the A.BY/SnJ murine strain (a susceptible strain) and TLR3-deficient mice indeed show lower CVB3-induced expression of interferon type I compared to the resistant C57BL/6 strain. Splenocytes of TLR3 knockout mice also showed reduced IFN-γ production [82]. In contrast, transgenic (over)expression of TLR3 in mice is sufficient to mount an effective immune response against CVB3, even in the absence of type I interferon signaling [189]. Interestingly, the enterovirus 3C protein is capable of cleaving TRIF in vitro, hereby inhibiting type I interferon production upon TLR3 stimulation. This could represent a sophisticated viral mechanism to dampen the antiviral effect of TLR activation [192].

Deficiency of TLR3 does not affect the maturation of antigen-presenting cells or the subsequent activation of the adaptive immune response. Adoptive transfer of wild type macrophages does reduce cardiac damage and mortality in TLR3-deficient mice [38].

Together, these findings suggest that TLR3 limits cardiac damage primarily by enabling the host to control viral replication.

### 8.3. TLR4

Less attention was paid to the role of TLR4 in viral myocarditis, presumably given the fact that TLR4 is expressed on the cell surface and only recognises distinct virus-specific components. TLR4 knockout resulted in increased coxsackievirus levels in the heart two days after inoculation. On the contrary, on day 12, viral presence was reduced, and less severe myocarditis was observed. These effects were attributed to decreased levels of the pro-inflammatory cytokines IL-1β and IL-18 in the heart [135]. The authors hypothetised that TLR4 signalling may protect the host against viral replication during the innate immune response, but may lead to autoimmune disease later on [154]. However, treating mice with LPS increased cardiac viral titers in both sexes, suggesting that TLR4 signalling is not beneficial during the viral replication stage *per se*. Treatment at the time of infection did not confer increased mortality. Treating female mice with LPS at day 3 postinfection did reduce survival [113].

### 8.4. TLR7 & TLR9

Pagni et al. made a head-to-head comparison of the effects of TLR3, TLR7 or TLR9 knockout in BALB/c mice inoculated with MCMV. While TLR3- and TLR7-deficient mice developed similar levels of myocardial inflammation compared to the wild-type controls, TLR9-deficient mice exhibited more severe myocarditis. These findings corresponded with the viral load being higher in the hearts of TLR9^-/-^ mice. No differences in cardiac troponin levels were observed when comparing the groups, not even in comparison with uninfected controls. This could be related to the time point of evaluation, which was in this case day 10 post-infection [148]. On the contrary, Riad et al. demonstrated that TLR9 deficiency was beneficial in the acute phase of CVB3-induced myocarditis in a resistant murine strain (C57BL/6). This effect was not related to cardiac viral load, but coincided with reduced immune cell infiltration and inflammatory cytokine levels [147].

### 8.5. MyD88

Mixed results were reported in animal studies concerning its role in myocarditis. In the trial by Fuse et al., inoculating C57BL/6 mice with coxsackievirus B3, MyD88-deficient mice showed decreased viral load in the myocardium (despite decreased type II interferon levels in heart and serum), less extensive myocarditis on histopathology and reduced mortality. The investigators observed reduced tissue CAR expression, which is in line with the aforementioned findings. Surprisingly, the levels of IRF-3 (including the activated form) and type I interferons were increased [69]. This could mean that the benefit of MyD88 knockout results from increased signalling through non-MyD88-dependent signalling pathways, raising type I interferon production.

In contrast, in the susceptible BALB/c strain, MyD88 deficiency increased viral load and inflammatory lesions in the myocardium [148]. In non-obese diabetic mice (NOD/ShiLtJ strain), no effects of MyD88 knockout were observed, contrary to TLR3 knockout, which had detrimental consequences [38].

### 8.6. TRIF

TLR3 and 4 are capable of signalling through a non-MyD88-dependent pathway which starts with the recruitment of TRIF. All studies were conducted in the C57BL/6 strain, and TRIF knockout resulted in increased viral presence, more extensive inflammatory lesions, reduced cardiac function and increased mortality. The effects on cardiac and serum cytokines were variable [171,189,190]. Riad et al. demonstrated a biphasic response of IFN-β gene expression with initially decreased levels. IFN-β treatment of TRIF-deficient mice improved virus control and reduced excess mortality [171].

**Figure 4 viruses-13-01003-f004:**
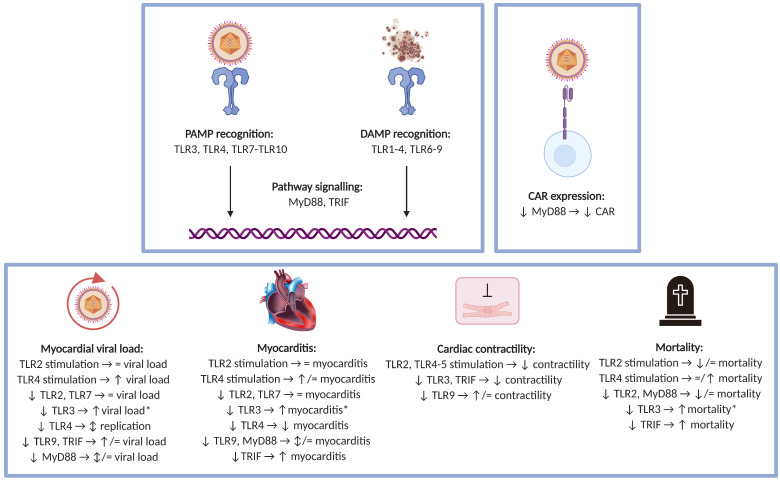
Proposed roles of individual Toll-like receptors in viral myocarditis. Based on the current evidence, the different TLRs take on different roles in the viral myocarditis disease process. The intracellular TLRs and TLR4 and TLR10 recognise virus-related PAMPs. The majority of TLRs are able to interact with DAMPs. MyD88 and TRIF are key molecules in the TLR signalling pathways. MyD88 influences CAR expression, which functions as a viral receptor. Deficiency or stimulation of individual TLRs or associated molecules differently impacts outcomes in the setting of viral myocarditis (myocardial viral load, myocardial lesions, cardiac contractility and mortality). * denotes generalisation based on the majority of findings reported in literature. Abbreviations: CAR, coxsackievirus-adenovirus receptor; DAMP, damage-associated molecular pattern; MyD88, myeloid differentiation primary response gene 88; PAMP, pathogen-associated molecular pattern; TLR, Toll-like receptor; TRIF, TIR-domain-containing adaptor protein-inducing interferon-β.

## 9. Cardiomyocytes in Viral Myocarditis and TLRs

Cardiomyocytes are an intricate cell type that constitute 90% of the adult heart’s mass [50,142,193]. Their regenerative capacity is negligible in mammals and the heart contains very few cardiomyocyte progenitor cells [76,142,162].

It turns out that the cardiomyocytes are more than passive bystanders/victims in the aforementioned pathogen-immune system interaction. Cardiomyocytes’ primary function is contraction, but it has been demonstrated that they also have fundamentally other properties. They can respond to danger signals with a complex inflammatory and functional response [42,43,145]. Therefore, a novel role for cardiomyocytes has been postulated, analogous in some respects to that of innate immune dendritic cells who have an initial response directed by TLRs. By signalling through TLRs, the cardiomyocyte inflammatory response involves cytokines (i.a. chemokines), with subsequent leukocyte recruitment and cell surface adhesion molecule expression. Cardiomyocytes themselves have been found to secrete cytokines and express surface immunologic molecules in response to systemic pro-inflammatory stimuli. The subsequent inflammatory response is strikingly similar to that seen with activation of professional immune cells [42].

While it is presumed that the primary purpose behind TLR-mediated production of cytokines (i.a. chemokines) and cell-surface adhesion molecules is the recruitment and activation of leukocytes, both groups of molecules also have important effects on cardiac contractility [194]. Stimulation of TLR2, TLR4 and TLR5 significantly reduces contractility of isolated murine ventricular cardiomyocytes [42]. Peptidoglycan-associated peptide (PAL) (a TLR2 agonist) reduces fractional shortening in mice and sarcomere shortening (and calcium transients) in isolated adult cardiomyocytes. Isolated cardiomyocytes from MyD88 or TLR2 knockout and TNFR1/2 knockout mice did not show effects on shortening and calcium cycling in response to PAL. Therefore, it was suggested that the effects were related to TLR2/MyD88 signalling with subsequent TNF-α release [195]. In vivo, administration of LPS dramatically decreased ejection fraction in wild type mice, while NF-kB knockout mice were not affected. Therefore, it seems that the downregulation of cardiomyocyte contractility potentially results from NF-kB dependent pathways [42]. We know that TLRs are important upstream activators of NF-kB signaling [16].

In vitro and animal research has demonstrated that pro-inflammatory cytokines (e.g., TNF-α, IL-1β, IL-2, IL-6, IFN-γ) are direct myocardial depressant factors [196,197,198,199,200,201,202,203,204,205,206]. IL-2 and IFN-γ have also been shown to cause myocardial depression in humans in vivo [207,208]. One of the mechanism underlying the decreased contractility could be outside-in signalling (transmission of signals from the outside of the cell to the inside, e.g., through ICAM-1) [42]. Simms and Walley demonstrated that direct contact with macrophages was required for rat ventricular myocytes to exhibit decreased fractional shortening upon challenge with LPS, TNF-α or IL-1. Anti-ICAM-1 prevented the decrease of fractional shortening [209]. In subsequent research, it was shown that cardiomyocyte ICAM-1 binding, by leukocytes or by other ligands, decreases cardiomyocyte contractility through alterations of the actin cytoskeleton [210].

## 10. Cardiac Fibroblasts in Viral Myocarditis and TLRs

Fibroblasts are one of the main stromal cell types in the healthy heart. They regulate extracellular matrix turnover and maintain tissue architecture [211].

In vitro work has shown that murine cardiac fibroblasts express high levels of CAR, similar to cardiomyocytes, and that expression is further increased 24 h after CVB3 infection. Therefore, it is not surprising that fibroblasts are infected with high efficiency by CVB3. They even display higher rates of virus replication compared to cardiomyocytes, potentially leading to aggravation of direct viral damage [212].

Next to their important role in extracellular matrix homeostasis, fibroblasts are increasingly recognised as key active player in the immune system. As already mentioned, fibroblasts express a range of TLRs and ligand activation of those receptors can lead to fibroblast activation with cytokine secretion, and promote differentiation into a myofibroblast phenotype [163,212]. Fibroblasts modify the quantity, quality, and duration of the inflammatory infiltrate and play a critical role in the transition from acute to chronic, persisting inflammation. Through release of chemokines, they drive homing of circulating leukocytes and by modulating adhesion molecule expression on endothelial cells, fibroblasts can promote or inhibit subsequent leukocyte recruitment [163,213]. Furthermore, myofibroblasts can signal leukocytes to produce matrix metalloproteinase 9 (MMP-9), which is not only the most relevant pro-fibrotic MMP, but also degrades the basal membrane, facilitating transendothelial migration. In addition, fibroblasts and myofibroblasts modulate the behaviour, retention/emigration and survival (i.a. apoptosis) of the recruited leukocytes. Dysregulation of the homeostatic balance between leukocyte recruitment, proliferation, emigration and survival may induce sustained inflammation [163].

## 11. TLR-Directed Therapy

TLRs fulfil many of the criteria to be considered potential therapeutic targets. Overexpression or knockout of TLRs and administration of ligands affect disease outcomes. Moreover, genetic differences (e.g., SNPs) in patients have been associated with disease risk. Cell surface TLRs can be targeted by small molecules (e.g., eritoran which inhibits TLR4) and antibodies (e.g., OPN-305 which targets TLR2). Intracellularly located TLRs require modified oligonucleotides to be targeted [98]. Rather than directly targeting TLRs, it may be more appropriate to modulate transcription regulation of TLRs (e.g., through inhibition of up- or downregulating cytokines) or translation (e.g., via microRNA-based therapy). The fact that TLR signalling converges to well-defined pathways opens opportunities to target downstream molecules [214]. For example, recently great therapeutic interest has been expressed in the Nod-like receptor pyrin domain containing 3 (NLRP3) inflammasome, an intracellular multimeric protein complex that activates inflammatory pathways. TLR-dependent NF-kB (activated through MyD88 or TRIF) and reactive oxygen species (ROS) appear to regulate both the priming and posttranslational steps required for assembly and activation of the inflammasome [215,216,217].

Drugs targeting TLRs (either as stand-alone or as adjuvant) are already being used in current clinical practice, including for the treatment of viral disease [98,214]. Imiquimod is a small-molecule TLR7 agonist that is approved for treatment of genital warts, a manifestation of HPV infection [32,98]. Heplisav B (FDA approved, EMA pending) is a recombinant hepatitis B vaccine combining hepatitis B surface antigen with the TLR9 stimulant CpG 1018 [98]. Many products targeting TLRs in the context of viral disease are currently under investigation [98,214].

TLR activation is an early step in the inflammatory cascade, but on the other hand, TLRs may be a key factor in the vicious cycle between inflammation and tissue damage [98]. This explains the significant interest in TLR antagonists for treatment of conditions in which the immune system is inappropriately overactive (e.g., autoimmune disease).

No aetiology-driven therapy is available for viral myocarditis. The protean nature of the disease hampers the search for a therapeutic breakthrough [218]. To the best of our knowledge, only one research group has administered pharmacological agents during viral myocarditis with the purpose to modulate TLRs (the results were ambiguous and seemed dependent on the sex of the animals as discussed earlier) [113].

Many questions surround therapies directed towards TLRs, both in general as in the context of viral myocarditis. For example, it remains unclear whether targeting a single TLR is sufficient or, given the importance of TLR crosstalk, several TLRs should be addressed simultaneously [32]. The profoundness of TLR modulation is likely to relate to development of adverse effects. Excessive or longstanding activation of the TLR pathways could give rise to perpetuating inflammation and fibrosis, and perhaps even autoimmunity [98]. We have emphasised the importance of TLRs in activating innate immunity and shaping adaptive immunity. Therefore, inhibiting one or more TLRs could negatively affect host immunity, a potential side effect of which the functional impact remains challenging to assess within the frame of animal experiments [32].

## 12. Limitations of Translational Viral Myocarditis Models

Whereas viral myocarditis has been experimentally induced in several animal species (hamster, rabbit, pig and monkey), systematic research on the role of TLRs is, to the best of our knowledge, limited to murine studies [19]. The mouse has become an invaluable model for viral myocarditis because of several reasons: genetic similarity to humans, comparable immune pathology and disease pattern to humans, strain-specific susceptibility, availability of transgenic strains and cost-efficient handling and breeding [6,56,218,219,220,221].

As already mentioned in the section on viral myocarditis, the CVB3-induced myocarditis model has provided most of our current insight into the disease [103,104]. However, from a translational point of view, this model is handicapped by the development of pancreatitis and a high systemic inflammatory response, which reflects more a CVB3 infection in infants rather than a disease in adults [4,221]. Furthermore, the fact that mainly CVB3 (and a few strains in particular) has been used, limits the generalisability [4,218].

Specifically for research involving TLRs, we must also take into account that during evolution, the demands of a rapidly changing (microbial) environment have imposed changes in coding and non-coding regions of the TLR genes. In addition, the expression and regulation at transcription level of TLRs differs between mice and humans, at least for several of the orthologues [222]. A last point of critique lies in the fact that, in general, translational models ignore the role of environmental factors [223].

## 13. Conclusions and Future Perspectives

Every phase of viral myocarditis has multiple points of connection with TLR function and signalling, although some still remain poorly studied at this time. TLRs take on a dual role during viral myocarditis as these connections include aspects where TLRs provide protection to the host as well as facets where their role is considered disadvantageous.

Viral myocarditis is a heterogeneous disease where the relative importance of each phase varies depending on host, pathogen, and their interaction. The role that TLRs play is also likely to vary according to these factors, which represents a plausible explanation for the ambiguity of some of the experimental results on the role of TLRs in viral myocarditis. These studies differ with regard to murine strain used (with associated intrinsic susceptibilities), virus administered and time point of evaluation. Future research on TLRs in viral myocarditis should explore the impact of different host strains and different viruses (with different mechanisms of cardiac injury), as this likely also reflects the situation in man. Although the subdivision of the disease process into distinct phases represents an artificial oversimplification, evaluating the impact of TLRs in each phase will provide more nuanced information on their role.

In general, TLR-knockout models are used to explore the role of TLRs, but this is a very dichotomous approach. More gradual and temporally confined TLR modulation could shed a better light on the complex pathophysiological interactions that TLRs are involved in.

In the clinical context, efforts are currently being made to study predisposing factors, pathobiology and long-term evolution of acute myocarditis via multicentre collaborations. These valuable initiatives should pay sufficient attention to differences in TLRs as determinants of the susceptibility and natural history. Better understanding of the TLR biology in the context of viral myocarditis may open new opportunities for evidence-based development of TLR-targeted therapy.

## Figures and Tables

**Figure 1 viruses-13-01003-f001:**
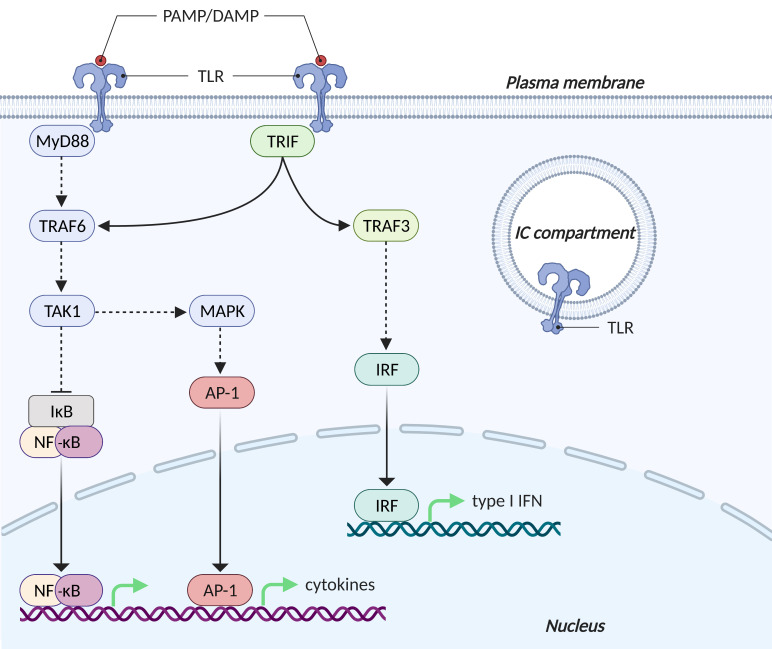
Simplified overview of the MyD88-dependent and -independent signalling pathways in Toll-like receptor activation. TLRs localise to the cell surface or intracellular compartments (such as endosomes, (endo)lysosomes and endoplasmic reticulum). Differential adaptor molecules initiate specific signalling pathways in different TLR family members. A detailed description of the pathways can be found in the text. Full lines indicate direct interaction. Dashed lines indicate indirect linkage. Abbreviations: AP-1, activator protein-1; IC, intracellular; DAMP, damage-associated molecular pattern; IFN, interferon; IRF, interferon regulatory transcription factor; MAPK, mitogen-activated protein kinase; MyD88, myeloid differentiation primary response gene 88; NF-κB, nuclear factor kappa B; PAMP, pathogen-associated molecular pattern; TAK, TGF-β-activating kinase; TLR, Toll-like receptor; TRAF, TNFR-associated factor; TRIF, TIR-domain-containing adaptor protein-inducing interferon-β.

**Figure 2 viruses-13-01003-f002:**
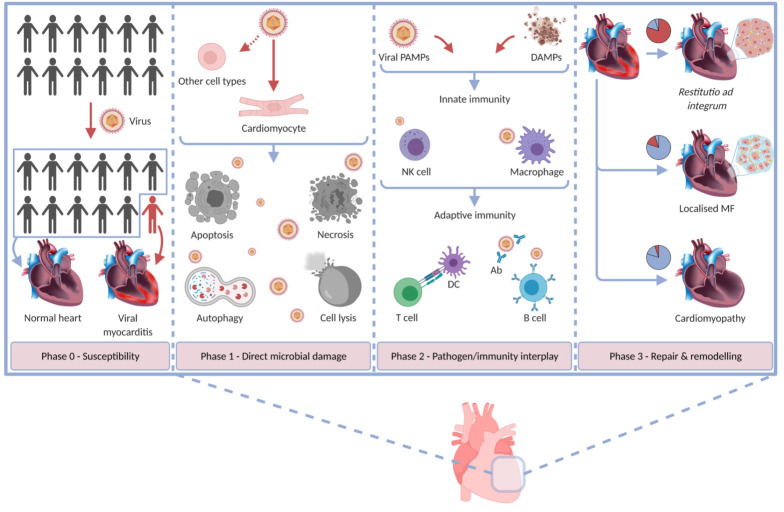
The different phases of viral myocarditis. The susceptibility phase refers to the predisposing factors which make that only a minority of virally infected individuals develops myocarditis. In phase I, the virus infects and replicates in the heart, causing tissue damage without involvement of the host immunity. The insulting pathogen and the tissue damage inflicted in phase I initiate a host immune response in phase II. At first, the innate immunity battles the intruding microbe, followed by an adaptive immune response. For the majority of patients, repair in phase III leads to complete recovery. However, in some, the tissue damage will result in localised scar tissue. A small fraction of patients develops cardiac dysfunction, with mostly a dilated phenotype in this context. Abbreviations: Ab, antibody; DAMPs, damage-associated molecular patterns; DC, dendritic cell; MF, myocardial fibrosis; NK, natural killer; PAMPs, pathogen-associated molecular patterns.

**Figure 3 viruses-13-01003-f003:**
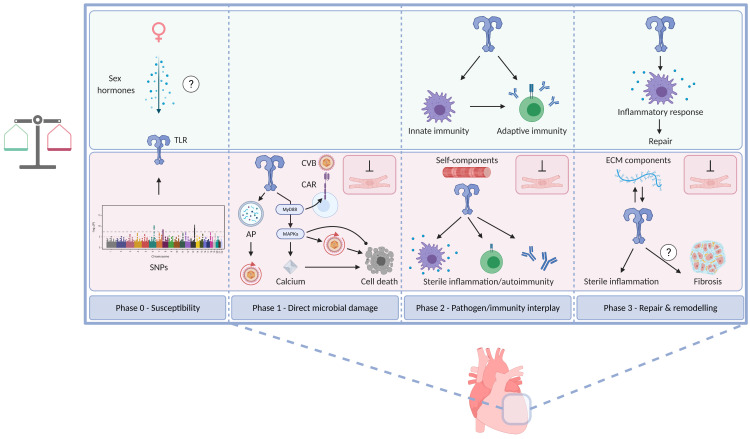
Proposed roles of Toll-like receptors in viral myocarditis. Every phase of the myocarditis disease process has several associations with Toll-like receptors (TLRs), although for some the evidence is limited. These include aspects where TLRs are considered beneficial (green plane) as well as elements where TLRs potentially confer harm to the host (red plane). The ratio between both aspects determines the overall impact of TLRs on the disease process, as indicated by the weighing balance. Phase 0: TLR sex differences have been put forward as explanation for the sex bias in viral myocarditis (in which the female sex offers protection). The impact of sex hormones could run through TLR modulation. Genetic differences in TLRs are among the factors that influence host susceptibility to viral myocarditis. Phase 1: TLRs can induce autophagy and viruses can use autophagosomes as a platform for their replication. MyD88 impacts expression of CAR, an important viral receptor. MAPK family members seem to have a role in CVB replication and apoptosis regulation. Activation of MAPK pathways leads to intracellular calcium mobilisation during viral replication, contributing to cell death. Phase 2: TLRs are central in innate and adaptive immune system activation in response to the offending virus. Inappropriate TLR activation by self-components propagates sterile inflammation and contributes to the development of autoimmunity. Phase 3: The inflammatory response promotes and coordinates the repair phase that follows the cardiac insult. A feedback loop between extracellular matrix (ECM) constituents and TLRs is suggested in which ECM molecules act as DAMPs and subsequent TLR activation affects the ECM. Besides sterile inflammation, ECM-mediated activation of TLRs could also cause other negative consequences, for example stimulation of fibrogenesis. In all phases, TLR activation can negatively affect cardiomyocyte contractility. Abbreviations: AP, autophagosome; CAR, coxsackievirus-adenovirus receptor; CVB, coxsackie B virus; ECM, extracellular matrix; MAPK, mitogen-activated protein kinase; MyD88, myeloid differentiation primary response gene 88; SNP, single-nucleotide polymorphism; TLR, Toll-like receptor.

**Table 1 viruses-13-01003-t001:** Studies investigating the role of TLRs in viral myocarditis.

First Author, Year (Ref)	Virus	Mice		Viral load		Histopathology: Inflammatory Lesions	Tn	Cytokine			Cardiac Function	Mortality (Length of FU)
			Comparator	Myocardial	Serum			Myocardial		Serum		
								mRNA	Protein			
TLR2												
Roberts, 2012 [113]	CVB3 (H3 variant)	♂ C57BL/6 + Pam3CSK4	♂ C57BL/6 + PBS	=		=						↓ (7 d)
		♀ C57BL/6 + Pam3CSK4	♀ C57BL/6 + PBS	=		=						= (7 d)
Roberts, 2013 [187]	CVB3 (H3 variant)	♂ TLR2^-/-^ (C57BL/6)	♂ C57BL/6	=		=						= (7 d)
		♀ TLR2^-/-^ (C57BL/6)	♀ C57BL/6	=		=						↓ (7 d)
TLR3												
Hardarson, 2007 [188]	EMCV	TLR3^-/-^ (W9.5)	TLR3+/+ (W9.5)	↑ (d3 & d5 PI)		↓ (d3 & d5 PI)	↑ (d3 PI)	TNF = d0 & d5 PI and ↓ d3 PI; IL-1β = d0 & d5 PI and ↓ d3 PI; IL-6 = d0 & d5 PI and ↓ d3 PI; IFN-β = d0 & d5 PI and ↑ d3 PI; RANTES = d0 PI and ↓ d3 & d5 PI; IP-10 = d0 & d5 PI and ↓ d3 PI; MIP-2, MIP-1α, MIP-1β = d0, d3, d5 PI				↑ (14 d)
		TLR3^+/-^ (W9.5)										↑ (14 d)
Negishi, 2008 [189]	CVB3 (Nancy strain)	Tlr3^-/-^ (C57BL/6)	C57BL/6	= d1 PI; ↑ d3, d6, d9 PI	= d1 PI; ↑ d2 & d3 PI	↑ (d12 PI)		IL-1β = d0, d1, d6 PI and ↓ d3 PI; IL-12p40 = d0 & d1 PI and ↓ d3 & d6 PI; IFN-γ ↓ d3 PI; IFN-β = d0, d1, d3, d6 PI				↑ (15 d)
		TLR3-Tg Ifnar1^-/-^	Ifnar1^-/-^		↓ (d2 PI)							↓ (7 d)
Weinzierl, 2008 [82]	CVB3 (Nancy strain)	TLR3^-/-^ (B6;129S1-Tlr3^tm1Flv^/J)	C57BL/6NCrl H-2^b^	↑ (d8 PI)		↑ (d8 PI)						
			A.By-H2^b^ H2-T18^f^/SnJ	↑ (d8 PI)		↑ (d8 PI)						
Richer, 2009 [38]	CVB4 (Edwards strain)	TLR3^-/-^ (NOD)	NOD/ShiLtJ	= d3 PI; ↑ d7 PI		↑ (d7 PI)	= d3 PI; ↑ d7 PI			IFN-α ↓ (d2 PI); TNF-α, CCL5 ↓ (d4 PI); IL-6, IFN-γ, CCL2, CCL3, CCL4, CXCL9 = (d4 PI)		↑ (21 d)
Pagni, 2010 [148]	MCMV (strain K181)	TLR3^-/-^ (BALB/c)	BALB/c	= (d10 PI)		= (d10 PI)	= (d10 PI)					
Abston, 2012 [190]	Heart-passaged CVB3 (Nancy strain)	TLR3^-/-^ (B6.129)	B6.129	↑ (d10 PI)		↑ (d10 PI)			IFN-γ ↓; IL-33, IFN-β =; IL-4 ↑ (d10 PI)		↓ (d10 PI)	= (35 d)
Sesti-Costa, 2017 [191]	CVB3 (Nancy strain)	TLR3^-/-^ (C57BL/6)	C57BL/6	= d1 & d3 PI; ↑ d12 PI		↑ (d12 PI)						↑ (25 d)
TLR4												
Fairweather, 2003 [135]	CVB3 (Nancy strain)	C3H/HeJ TLR4^-/-^ (missense mutation which prevents functional TLR4 signalling)(BALB/c)	BALB/c	↑ d2 PI; ↓ d12 PI		↓ (d12 PI)			IL-1β, IL-18 ↓; IL-12p70, TNF-α, IFN-γ = (d12 PI)			
Roberts, 2012 [113]	CVB3 (H3 variant)	♂ C57BL/6 + LPS	♂ C57BL/6 + PBS	↑		↑						= (7 d)
		♀ C57BL/6 + LPS	♀ C57BL/6 + PBS	↑		=						= (treatment at day of infection) (7 d); ↑ (treatment at d3 PI) (7 d)
TLR7												
Pagni, 2010 [148]	MCMV (strain K181)	TLR7^-/-^ (BALB/c)	BALB/c	= (d10 PI)		= (d10 PI)	= (d10 PI)					
TLR9												
Pagni, 2010 [148]	MCMV (strain K181)	TLR9^-/-^ (BALB/c)	BALB/c	↑ (d10 PI)		↑ (d10 PI)	= (d10 PI)					
Riad, 2010 [147]	CVB3 (Nancy strain)	TLR9-/-(C57BL/6)	C57BL/6	= (d7 & d28 PI)		↓ d7 PI; = d28 PI		TGF-β ↓ d7 PI and = d28 PI; IFN-β ↑ d7 PI and = d28 PI	TNF-α ↓ d7 PI; = d28 PI		↑ d7 PI; = d28 PI	
MyD88												
Fuse, 2005 [69]	CVB3 (cardiovirulent strain)	MyD88^-/-^ (C57BL/6J)	C57BL/6	↓ d4, d7, d10 PI; = d14 PI		↓ (d4, d7, d10, d14 PI)		IL-1β, IL-10, IL-18, TNF-α, IFN-α, IFN-β, IFN-γ = d0 PI; IL-1β, IL-18, TNF-α ↓ d4, d7, d10 PI; IL-10 = d4 & d10 PI and ↓ d7 PI; IFN-α ↑ d4 PI and = d7 & d10 PI; IFN-β ↑ d4 & d7 PI and = d10 PI; IFN-γ ↓ d4 & d7 PI and = d10 PI		IL-1β, IL-2, IL-6, IL-12, TNF-α, IFN-γ ↓ d7 PI and = d0, d10, d14 PI; IL-4, IL-10 = d4, d7, d10, d14 PI		↓ (14 d)
Richer, 2009 [38]	CVB4 (Edwards strain)	MyD88^-/-^ (NOD)	NOD/ShiLtJ	= (d3 & d7 PI)		= (d7 PI)	= (d3 & d7 PI)			IFN-α ↓ (d2 PI); TNF-α, CCL2, CCL3, CCL4, CCl5, CXCL9 = (d4 PI); IL-6, IFN-γ ↓ (d4 PI)		= (21 d)
Pagni, 2010 [148]	MCMV (strain K181)	MyD88^-/-^ (BALB/c)	BALB/c	↑ (d10 PI)		↑ (d10 PI)	= (d10 PI)					
TRIF												
Negishi, 2008 [189]	CVB3 (Nancy strain)	Trif^-/-^(C57BL/6)	C57BL/6		↑ (d2 PI)			IL-12p40, IFN-γ ↓; IFN-β = (d3 PI)				
Riad, 2009 [171]	CVB3 (Nancy strain)	Trif^-/-^(C57BL/6)	C57BL/6	= 12h & 24h PI; ↑ d2 & d7 PI		↑		IL-1β, IL-10, IL-18, TNF-α, IFN-β = 12h PI; IL-1β, IL-10, TNF-α, IFN-β = 22h PI; IL-18 ↓ 24h PI; IL-1β, IL-10, IL-18, TNF-α = 48h PI; IL-1β = 72h PI; IL-10, IL-18 ↑ 72h PI; TNF-α, IFN-β ↓ 72h PI; IL-1β, IL-10, IL-18, TNF-α, IFN-β ↑ d7 PI	IL18, TNF-α = d3 PI; IL-1β ↑ d3 PI; IL-1β, IL-18, TNF-α ↑ d7 PI		↓ (d7 PI)	↑ (70 d)
Abston, 2012 [190]	Heart-passaged CVB3 (Nancy strain)	Trif^-/-^ (C57BL/6J)	C57BL/6J	↑ (d10 PI)		↑ (d10 PI)			IFN-β ↓; IFN-γ, IL-4 =; IL-33 ↑ (d10 PI)		↓ (d10 & d35 PI)	↑ (35 d)
All studies used intraperitoneal injection for virus inoculation. All research groups used haematoxylin and eosin staining for pathological assessment, except for Negishi et al. who used the Masson’s trichrome stain. Histopathological assessment was based on the extensiveness of inflammation/cellular infiltration. In the study by Fuse et al., also necrosis was part of the assessment. In the publication by Richer et al., the scoring was not specified. Abbreviations: CVB, coxsackievirus group B; d, day; EMCV, encephalomyocarditis virus; FU, follow-up; IFN, interferon; IL, interleukin; IP, inducible protein; MCMV, murine cytomegalovirus; MIP, macrophage inflammatory protein; mRNA, messenger RNA; PI, post infection; TNF, tumor necrosis factor; TRIF, TIR-domain-containing adapter-inducing interferon-β.												

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
