# Peer review of "Toll-Like Receptors: Are They Taking a Toll on the Heart in Viral Myocarditis?"

_viruses, 2021, doi:10.3390/v13061003_

Round 1
Reviewer 1 Report
This manuscript attempts to provide the overview of associations between TLRs and viral myocarditis. Infection by various viruses such as CVB3 may result in clinical and subclinical myocarditis. The knowledge regarding to the role of TLR signaling in viral myocarditis may provide hints to develop novel therapeutic strategies for treating myocarditis patients. This paper is well organized and provide abundant information. However, it will be beneficial if the authors can provide information regarding the TLR expression levels in rodents and compare the roles of TLRs playing in murine models and in humans.
- Lane 564, please correct the sentence.
- Please provide appropriate reference for “Detection of SARS-CoV-2 viral particles in cardiac macrophages implies that 186 the virus can reach the heart during viraemia of through infiltration of infected macrophages”
- Autophagy has been demonstrated to play an essential role in cardiomyopathy and viruses are with the ability to induce autophagy via TLRs. It will make this article more complete if the authors can discuss this part.
Author Response
Dear Sir/Madame
Thank you for reviewing our manuscript.
Please find our point-by-point responses to your comments in the file in attachment. We have asked the assigned assistant editor to also provide to you the manuscript with track changes.
Yours sincerely,
Kasper Favere

Reviewer 2 Report
The authors did an excellent job of reviewing both the biology of Toll-Like Receptors and their potential role in viral myocarditis. The manuscript will provide substantial valuable information for the reader in this area. If I might make a few very minor comments, while it is true that most TLR are expressed on immune cells, not all cells of the immune system express the same set of TLR. This basically means that different immune cells can respond differently including with production of distinct cytokines to TLR signaling (various dendritic cell populations for example). Also, classically, TLR2 activation most often is associate with immunosuppression/Treg cells whereas TLR4 activation is most often associated with immunopathogenicity so if there are differences in infiltrating cells expressing either TLR in the heart, this would bias the response one might assume. Also, although I might have missed it, I didn't see Madeleine Cunningham's paper reporting that cardiac myosin is a TLR2 agonist discussed. The authors might disagree but in considering TLR in viral myocarditis, I would think that cardiac myosin as an agonist would be relevant to possible pathogenesis. Finally, as the authors state, why some individuals develop clinical myocarditis while other do not is likely complex. However, it is probably important to remember that the clinical disease is the tip of the iceberg. Early sequential autopsy studies found fairly high incidences of cardiac inflammation which they designated "myocarditis" and which would not have been seen as clinical myocarditis. There were abnormal EKG in 5-12% of patients with active flu. These have always been controversial as cytokines can impact cardiac function and the definition of myocarditis has been debated for decades. Nonetheless, the fact that individuals die of sudden unexpected death due to myocarditis should indicate that there must be unsuspected myocarditis in the population - we simply don't know how much.
Cutting edge: cardiac myosin activates innate immune responses through TLRs.
J Immunol. 2009 Jul 1;183(1):27-31. Cardiac myosin-Th17 responses promote heart failure in human myocarditis. JCI Insight. 2016 Jun 16;1(9):e85851.Author Response
Dear Sir/Madame
Thank you for reviewing our manuscript.
Please find our point-by-point responses to your comments in the file in attachment. We have asked the assigned assistant editor to also provide to you the manuscript with track changes.
Yours sincerely,
Kasper Favere
